# Together in Care: An Enhanced Meals on Wheels Intervention Designed to Reduce Rehospitalizations among Older Adults with Cardiopulmonary Disease—Preliminary Findings

**DOI:** 10.3390/ijerph19010458

**Published:** 2022-01-01

**Authors:** Panagis Galiatsatos, Adejoke Ajayi, Joyce Maygers, Stephanie Archer Smith, Lucy Theilheimer, Sherita H. Golden, Richard G. Bennett, William Daniel Hale

**Affiliations:** 1Office of Diversity and Inclusion, Johns Hopkins Health System, Baltimore, MD 21205, USA; sahill@jhmi.edu; 2Department of Medicine, Medicine for the Greater Good, Johns Hopkins School of Medicine, Baltimore, MD 21224, USA; ajayi@mowcm.org (A.A.); whale5@jhmi.edu (W.D.H.); 3Division of Pulmonary and Critical Care Medicine, Department of Medicine, Johns Hopkins School of Medicine, Baltimore, MD 21205, USA; 4Meals on Wheels, Baltimore, MD 21224, USA; archersmith@mowcm.org (S.A.S.); lucy@mealsonwheelsamerica.org (L.T.); 5Office of Population Health, Johns Hopkins Bayview Medical Center, Baltimore, MD 21224, USA; jmaygers@jhmi.edu; 6Departments of Medicine and Epidemiology, Johns Hopkins University, Baltimore, MD 21205, USA; 7Division of Geriatrics, Department of Medicine, Johns Hopkins School of Medicine, Baltimore, MD 21224, USA; rbennett@jhmi.edu; 8Office of the President, Johns Hopkins Bayview Medical Center, Baltimore, MD 21224, USA

**Keywords:** community health, rehospitalizations, COPD, health disparities

## Abstract

Rehospitalizations in the Medicare population may be influenced by many diverse social factors, such as, but not limited to, access to food, social isolation, and housing safety. Rehospitalizations result in significant cost in this population, with an expected increase as Medicare enrollment grows. We designed a pilot study based upon a partnership between a hospital and a local Meals on Wheels agency to support patients following an incident hospitalization to assess impact on hospital utilization. Patients from an urban medical center who were 60 years or older, had a prior hospitalization in the past 12 months, and had a diagnosis of diabetes, hypertension, heart failure, and/or chronic obstructive pulmonary disease were recruited. Meals on Wheels provided interventions over 3 months of the patient’s transition to home: food delivery, home safety inspection, social engagement, and medical supply allocation. Primary outcome was reduction of hospital expenditure. In regard to the results, 84 participants were included in the pilot cohort, with the majority (54) having COPD. Mean age was 74.9 ± 10.5 years; 33 (39.3%) were female; 62 (73.8%) resided in extreme socioeconomically disadvantaged neighborhoods. Total hospital expenditures while the cohort was enrolled in the transition program were $435,258 ± 113,423, a decrease as compared to $1,445,637 ± 325,433 (*p* < 0.01) of the cohort’s cost during the three months prior to enrollment. In conclusion, the initiative for patients with advanced chronic diseases resulted in a significant reduction of hospitalization expenditures. Further investigations are necessary to define the impact of this intervention on a larger cohort of patients as well as its generalizability across diverse geographic regions.

## 1. Introduction

Over the last several decades, the burden of disease has shifted from acute care toward chronic illness management [1]. The impact of this shift is seen in healthcare costs, which have continued to rise over recent years, and in medical care, which is becoming more complex, especially for older adults [2,3,4,5]. Older adults have an increased need for health information and resources to manage their health, health services, and overall wellness. However, national literacy surveys indicate that more than 70% of adults 65 or older in North America lack basic health literacy skills needed for successful interactions with the health system and its services and support [6]. Navigating through complex medical systems along with achieving medication compliance and lifestyle changes is challenging for a significant percentage of older adults, culminating in poor health outcomes and costly health care. Therefore, cost-effective methods to enhance current health care services for older adults must be a priority for health care systems.

In October 2012, the Centers for Medicare & Medicaid Services began financially penalizing hospitals with excessive risk-standardized readmission in an effort to reduce Medicare member rehospitalization rates and costs [7]. This policy led hospitals to develop programs to reduce readmission rates for patients. Many hospitals and systems developed disease-specific readmission reduction strategies for populations long-known to be at high risk for readmission (e.g., heart failure, chronic obstructive pulmonary disease, diabetes), and others developed transition-of-care interventions for patients discharged from the hospital with high disease burdens. Typically, these programs provided telephonic management and/or health care worker home visits. One type of intervention designed to prevent readmissions among populations transitioning from one care setting to another is often called “transitional care interventions” [8,9,10]. These aim to avoid unwanted outcomes, such as preventable readmissions, caused by uncoordinated care [8]. While there are no established components that best define what a transitional care intervention program should be, usually they involve focus on education (patient and/or caregiver), medication compliance and reconciliation, and coordination among health professionals involved in the transition team.

We designed a new approach to care transition team through the partnership of a hospital and a community organization (Meals on Wheels) in order to assist older adults returning home from the hospital. We hypothesized that this transition team would result in a reduction of hospital expenditure over the first 30-day period of a hospital discharge. The secondary outcome was focused on the impact on chronic obstructive pulmonary disease (COPD), a common cause of hospitalizations of older adults in our hospital.

## 2. Methods

### 2.1. Patient Population

We recruited patients at an urban, academic hospital—Johns Hopkins Bayview Medical Center, Baltimore City, Maryland. These patients were selected if they were ≥60 years old and had had a prior hospitalization in the past 12 months or 2 prior emergency department visits in the past 12 months. Further, patients had primary admissions based on one of four common diseases that result in frequent rehospitalizations [11]: heart failure, diabetes, hypertension, and/or chronic obstructive pulmonary disease. Once identified, patients were approached at least 48 h prior to the expected discharge to discuss enrolling in our initiative. Exclusion criteria included non-English-speaking patients, those without a permanent home for 3 months following hospital discharge, and the inability to consent. The study was approved by the institutional review board (IRB) at Johns Hopkins School of Medicine (Study Number CR0034409, IRB00175698). Written informed consent was obtained from the participants or from the participant’s next of kin in accordance with the Helsinki II declaration.

### 2.2. Together in Care Initiative Details

#### 2.2.1. The Team

The initiative, called “Together in Care”, was a partnership between an academic, urban hospital and the Meals on Wheels of Central Maryland (MOWCM). We created a transition team for patients who agreed to enroll in the effort to reduce 30-day readmissions. If a patient agreed to enroll into the transition program, MOWCM was notified immediately after the patient’s consent. Further, at the time of their consent, patients identified their primary health care professionals. We then contacted and notified them of their patient’s enrollment into this transition-of-care program. Finally, the patient’s post-hospitalization medical plan was reviewed by study team members and MOWCM IRB-approved staff in an effort to assure the survey questions, discussed below, included pertinent information that the hospital team believed would allow for the patient to recover at home.

MOWCM provided two personnel to help with the patient’s transition. First, there was a dedicated volunteer who would deliver food items and execute the daily survey and social engagement. The volunteer read the questions from the survey to the participant. This volunteer would record all responses in real time via an electronic tablet, where responses then would be immediately available to be reviewed by the MOWCM care manager. Note that the dedicated volunteer maintained the same participants during the trial, as consistency in the volunteer for the participant was viewed as an advantage for improving social engagement. Second, the care manager from MOWCM was vital to coordinating care for the participant as questions were answered via the survey on the electronic tablet. The care manager would then relay concerning responses (e.g., “I am not taking my medications”) to the patient-identified clinician for further management. The care manager was also responsible for assuring medical supply allocation occurred (e.g., a scale to manage heart failure, finger sticks for diabetes management).

#### 2.2.2. The Interventions

MOWCM provided four interventions during the patient’s transition to home. First, MOWCM delivered food that was in compliance with the American Heart Association and American Diabetes Association standards. These meals were delivered daily by a volunteer or staff member, with enough meals delivered on Friday to last the weekend. Note, during the delivery of meals, each MOWCM personnel spent up to 10 min with the patient, asking questions from a predetermined survey regarding the participant’s current symptoms, mood, and compliance with medical advice post-hospitalization.

Second, MOWCM performed a home safety inspection to identify home concerns (e.g., fall hazards, running water, electricity). Minor home repairs such as lighting improvement, shower handrails, and other fall reduction improvements were implemented in order to create a safe home environment that allows for maximum mobility. Modifications to the home were made if the participant agreed to them (e.g., adding grab handles to the wall) within the first 2 weeks of returning from the hospital.

Third, the MOWCM care manager discussed with the patient any medical supplies identified by the discharging medical providers at the hospital that the patient would benefit from during their transition home (e.g., blood pressure cuff, cane). Any identified equipment was obtained by the MOWCM care manager and delivered to the participant.

Finally, during the delivery of every meal, social engagement by the MOWCM volunteer was coordinated through a scripted daily questionnaire for each participant, as mentioned above. The questionnaire consisted of questions about compliance with medications and attending post-hospital health appointments, as well as overall insight into mood, appetite, and well-being. The questionnaire was typically completed in under 10 min, with the questions logged into an electronic tablet and relayed in real time to the MOWCM care manager through a secure network.

The MOWCM care manager identified any worrisome responses and notified the patient’s primary health team (identified prior to hospital discharge by the enrolled patient). Such concerns, for instance, ranged from medication non-compliance to the patient reporting a loss of appetite. All enrolled participants had their meals delivered and daily surveys completed by noon of each weekday, with results relayed to health care professional offices no later than 1 pm that business day by either fax or telephone call. Further, the MOWCM care manager discussed with the hospital research partners any concerns they encountered during the collection of the surveys and the responses.

This service was provided for 3 months, Mondays through Fridays (off for weekends and federal holidays) and did not interfere with standard care for patients discharged from the hospital. Note on Fridays, the participants received enough meals to last through the weekend. The participants did have the ability to notify the MOWCM care manager of any concerns over the weekend as well. Enrolled participants also had the option at the end of the 3-month trial period to retain certain standard Meals on Wheels services (e.g., usual food and nutrition delivery).

### 2.3. Variables Collected 

Demographic variables (age, sex, race, ethnicity, and ZIP code) were collected. Patient’s co-morbidities as well as medications and past medical history were also collected. The past medical history included information from the hospitalization during which the patient was enrolled, as well as their prior hospitalizations and/or emergency department visits: length of stay, intensive care unit utilization and length of stay if appropriate, and post-discharge plan (home versus sub-acute rehabilitation). Further, information on each participant’s hospitalizations after enrollment was documented during the 3-month intervention, then at 6 months and 12 months.

Expenditures specific to hospitalizations (total cost of care of hospital spending) were captured for patients up to 12 months after enrollment into the transition of care program, as well as 12 months prior to enrollment. Comparisons of these hospital costs were made between “3 months before” versus “first 3 months”, “6 months before” versus “first 6 months”, and “12 months before” versus “first 12 months”. Hospitalization expenditures were extracted from payer claims data available in the Chesapeake Regional Information System for our Patients (CRISP) [12]. These claims data were for all hospitals the patients may have been readmitted to before enrollment and during enrollment of Together in Care. 

Several variables were calculated based on the collected factors. These included sequential organ failure assessment upon admission to the hospital during which the patient was recruited, Charlson Co-morbidity index [13,14,15], and the area deprivation index (ADI) [11,16] to capture the socioeconomic status of each patient’s respective census tract neighborhood (the higher the score, the more disadvantaged the neighborhood). Note that the ADI was further evaluated as a categorical variable for individuals having an ADI of 85 or greater, representing living in the most disadvantaged neighborhoods [11].

### 2.4. Outcomes

The primary goal of this pilot trial was to evaluate the feasibility of the intervention. Therefore, we opted to recruit 100 patients over a 12-month period to assess feasibility and process outcomes of the pilot trial. The primary outcome for the pilot focused on hospital expenditures. A subgroup analysis focused on outcomes in patients with COPD, as COPD-related readmissions was the top cause of readmissions in the preceding year. Secondary outcomes included readmissions to the hospital or the emergency department (within 30 days, 3 months, and 6 months). Further, we recorded responses on the surveys, conducted by the personnel who delivered the food, that were believed to be identified as barriers to health and health care. 

### 2.5. Analysis

Where appropriate, mean ± standard deviation is provided for continuous variables and percentage for categorical variables. Pre- and post-hospital expenditures were compared by a two-sample *t*-test. A Kaplan–Meier curve was produced for readmissions over a 90-day period for the participants. All data analyses were conducted with R software (version 0.99.903, R Studio, Inc, Boston, MA, USA).

## 3. Results

Over a 12-month period (January 2019 to December 2019), 107 patients were enrolled into the program. Of the 107 patients who consented, 23 patients terminated their enrollment early (before the 3-month end date). The main reasons for termination included inconsistent home presence (16), issues with the food (5), and self-termination for undisclosed reasons (2). Therefore, the evaluation of the pilot study moving forward includes our 84 patients who completed the study.

### 3.1. Baseline Characteristics

The mean age of the participants was 74.9 ± 10.5 (range 60 to 93) years, with 33 (39.3%) being female. The mean body mass index of the patients was 27.0 ± 6.3. In regard to the socioeconomic status, the majority of patients resided in disadvantaged neighborhoods, as identified by the area deprivation index, with a mean 78.2 ± 17.2 (range 52 to 99). A complete listing of the cohort’s sociodemographic variables can be found in Table 1. 

In regard to the cohort’s hospitalization that resulted in enrollment into the trial, the mean length of hospitalization was 6.5 ± 4.6 days (range 1 day to 13 days), and 41 (48.1%) participants had an intensive care unit admission as well. Of note, 80 (95.2%) participants had an intensive care unit admission within the past 12 months. The Charlson Co-Morbidity Index of the cohort was 6.6 ± 2.5 (range 3 to 11), whereby the median 10-year survival of the cohort was 2% (IQR 0%, 21%) [15]. The most common co-morbidity amongst the cohort was hypertension, reported for 72 patients (85.7%). The most common co-morbidity that resulted in hospitalization was COPD, affecting 54 patients (64.3%). Finally, the mean number of medications for the cohort upon returning home from the hospital was 15.1 ± 6.6 (range 5 to 25), with the most medications for patients with COPD, 19.0 ± 7.1 (range 9 to 32). Table 1 further summarizes other medical information of the cohort.

### 3.2. Feasibility and Process Outcomes

Each participant in the transition team received home inspections, food deliveries, and daily surveys. Through the surveys, and in discussions with our care manager, several issues were identified that were not reported in hospital in-patient notes and discharge paperwork. Illiteracy was identified in two patients, while overall health literacy [6,17,18] concerns were identified in all of the participants. Through the surveys, several key issues were identified by the participants, ranging from an ongoing desire to be independent to a general anxiety over attending physician clinical visits, to feeling that health issues are not a priority compared to managing economic issues (e.g., employment, housing affordability). Table 2 summarizes key themes that were identified by our transition team that were felt to impact the overall management of the participant’s co-morbidities, with various example summaries from the surveys. Note these responses were grouped into six themes: “Insight into Health Condition”, “Trust of the Medical System”, “Access to Health Care Equipment”, “Competing Priorities”, “Caregiver Concerns”, and “Communication Access”. Medical supplies (e.g., blood pressure cuffs, scales) were purchased for 14 participants through MOWCM. Of note, no home falls were reported by the patients after their home renovations were completed.

### 3.3. Expenditure Outcomes and Subgroup COPD Analysis

Of the 84 participants enrolled, 49 were followed for 12 months and had their respective cost savings in regard to hospitalizations after enrollment in Together In Care analyzed. At the 3-month mark of enrollment, of the 49 participants, 48 were hospitalized, with total hospital expenditures being $435,258 ± 113,423 as compared to $1,445,637 ± 325,433 (*p* < 0.01) of their cost during the 3 months prior to enrollment (Figure 1). At the 6-month mark of enrollment, of the 49 participants, 42 experienced hospitalizations between months 3 to 6, with total hospital expenditures (all 6 months) being $899,106 ± 243,987 as compared to $1,853,406 ± 477,586 (*p* < 0.01) of their cost during the 6 months prior to enrollment. At the 12-month mark of enrollment, of the 49 participants, 17 experience hospitalizations between months 6 to 12, with total hospital expenditures (all 12 months) being $901,382 ± 375,588 as compared to $1,575,947 ± 273,444 (*p* = 0.04) of their cost during the 12 months prior to enrollment (Figure 1).

Of the 49 participants, 30 (61.2%) had COPD as a co-morbidity that was the primary diagnosis of rehospitalization. At the 3-month mark of enrollment, of the 30 participants, 17 were hospitalized with total hospital expenditures being $102,111 ± 12,573 as compared to $845,637 ± 207,587 (*p* < 0.01) of their cost during the 3 months prior to enrollment. At the 6-month mark of enrollment, of the 30 participants, 20 experienced hospitalizations between months 3 to 6, with total hospital expenditures (all 6 months) being $425,633 ± 43,782 as compared to $1,555,227 ± 122,685 (*p* < 0.01) of their cost during the 6 months prior to enrollment. At the 12-month mark of enrollment, of the 30 participants, 11 experienced hospitalizations between months 6 to 12, with total hospital expenditures (all 12 months) being $702,917 ± 210,325 as compared to $925,782 ± 355,286 (*p* = 0.03) of their cost during the 12 months prior to enrollment. The reduction in expenditures was due to significant reduction in intensive care unit, from 28 intensive care unit admissions for COPD-related disease to 0 intensive care unit admissions after enrollment.

### 3.4. Rehospitalization

Twenty-seven (32.1%) patients were readmitted within 30 days (Figure 2. Of those 27, only one was readmitted to an intensive care unit. The majority of the participants’ (20 of the 27) readmissions occurred within the first 15 days after returning from the hospital. Over the 3-month period, 37 total patients were readmitted to the hospital, and this cohort of 37 participants had an additional 122 emergency room visits during the 3-month span that did not result in hospitalization, resulting in 159 total hospital visits. In comparison to the 3-months prior enrollment of the complete cohort, there were 181 hospital visits: 142 admissions and 41 emergency room visits that did not result in hospitalizations. Comparing hospitalizations before and after enrollment over 3 months, there was a significant reduction in hospital admissions (142 before enrollment versus 37 after enrollment, *p* < 0.001), while there was an increase in emergency room visits that did not result in admissions (41 emergency room visits versus 122 emergency room visits, *p* < 0.001). Ten patients passed away within 3 months of enrolling into the program. 

## 4. Discussion

In the pilot trial of our Together in Care transition team initiative, we were able to demonstrate feasibility of the project that resulted in reduction of hospital expenditures for patients with advanced chronic diseases. The participants who were included in our cohort—recruited based on age and frequency of prior 12-month hospitalization and ED utilization—were challenged from both a medical standpoint (given their advanced Charlson Co-Morbidity) and a socioeconomic standpoint (given their neighborhood’s severe area deprivation index). Further investigations are warranted to establish the cost benefit of caring for elderly patients with advanced disease and significant socioeconomic disadvantage.

Reduction of hospital expenditures likely occurred through a multifactorial approach to the delivery of health-related items to our patients in this pilot. In addition to food delivery, a service well established with Meals on Wheels, several other interventions occurred through our Together in Care project. For instance, a focus on home safety resulted in modest home repairs for all of our participants. Falls at home for persons 65 and older occur frequently after hospitalizations and are a significant source of emergency room visits and hospitalizations, and home inspection interventions to prevent falls have resulted in significant fall reduction among older patients [19,20,21]. So, our home safety inspections focused on (1) access to electricity, water; (2) ease of mobility throughout the home; (3) minor interventions such as the installation of grab bars for safety. Overall, executing a home safety inspection with renovations was feasible during this pilot trial, and no falls were recorded for any of our participants during the study design. 

The next target area was social isolation, which of itself is a significant health and wellbeing issue for older persons living in the community. Social isolation has been associated with all-cause mortality, increased risk for re-hospitalizations, and increased falls [22,23,24]. In an effort to provide daily insight into how patients were adjusting to their post-hospitalization medical management, we extended the time allowed for meal delivery in Together in Care. Rather than 2-minute drop-off times, each food deliverer spent 10 min or more with the participant. During these visits, deliverers were scripted to do daily surveys of mood, appetite, and medication compliance. These conversations also allowed for social engagement between the participant and the food deliverers, social engagements that are likely to have significant positive impacts on health outcomes as previously reported [25,26]. In addition to the data collection, a goal of the project was to use assigned consistent personnel for each participant in our study’s cohort. This study design element was feasible and made an impact on our ability to monitor medical compliance and adherence, which in turn, likely contributed to the observed reduction in re-hospitalizations. This means that expanding the engagement time between a transition team’s personnel and the participant, along with a predetermined script for the consistent personnel for each participant, led to a meaningful collection of health-related symptoms and medical adherence data that resulted in mitigation of any severe exacerbations of the participant’s underlining medical condition.

Finally, the delivery of food, home improvements, and social engagement was well received by the recruited participants, who resided in significantly socioeconomically disadvantaged neighborhoods. The socioeconomic status of a neighborhood has been shown to impact hospital admissions and readmissions, especially for patients with COPD [11,27,28]. With this insight, we now recognize more fully that some transition team impact, especially if only focused on medical management, may be significantly attenuated, if not entirely undone, by other factors patients face in a socioeconomically disadvantaged neighborhood (e.g., lack of access to nutritious food, social isolation, and unsafe homes). Addressing these “non-medical” variables, as demonstrated in our pilot, may certainly have contributed to the observed clinical and economic impact. Specifically, in our COPD cohort, we saw no intensive care unit admissions once patients were enrolled into our program. Such a finding warrants further investigation, especially for such patients living in socioeconomically disadvantaged neighborhoods.

Several limitations were identified during the pilot that we did not specifically address. First, given the severity of co-morbidities in our participants, with many having end-stage organ dysfunction, several participants died within the 6-month period. Thereby, an additional outcome to be added in future transition-of-care initiatives is consideration and initiation of palliative care as appropriate for the participants. Second, assessment of quality of life is warranted. This will assure that clinical and economic outcomes are matched by participants discussing what enrollment in this transition team means for their well-being and fulfillment. Finally, our intervention was based on patients residing in an urban and suburban community, served by an urban, academic medical center. Evaluating such transition teams from different health care organizations and in other environments (e.g., rural, non-academic) is necessary to understand how such transition interventions can be adapted to diverse patients and communities. However, given these limitations, the pilot study holds significant potentials for at-risk older patients returning from the hospital in regard to mitigating further health decline through food access, home safety inspections, social engagement, and daily review of symptoms and medical advice adherence. 

In our Together in Care pilot, we found that a transition team focused on nutrition, home renovations, and social engagement resulted in a reduction of re-hospitalizations and overall hospital cost for enrolled participants. Our preliminary results affirmed the feasibility of the study design demonstrated significant impact on clinical and economic endpoints. Utilizing interventions that alleviate health insecurities of older, aging patients, while reducing their significant costs (e.g., long hospitalizations) should be a goal for all health systems. Our pilot, through a project between a community partner and a hospital, offers guidance for such an achievement. While a larger trial is warranted to evaluate this strategy against current transition-of-care management programs for elderly patients, we believe health systems overall should invest in such transitions of care.

## Figures and Tables

**Figure 1 ijerph-19-00458-f001:**
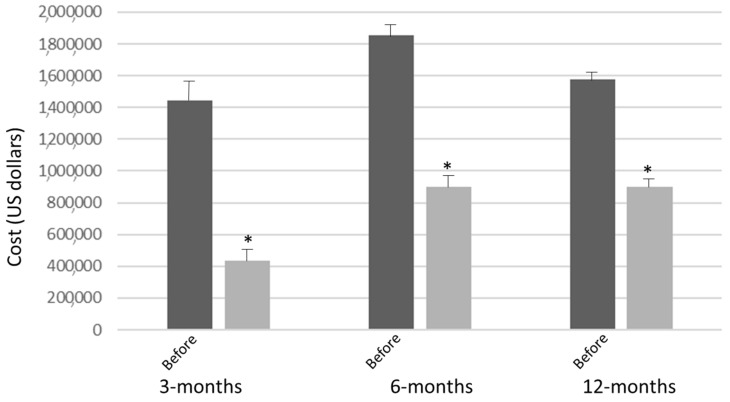
Hospitalization cost, comparing the immediate months prior to enrollment in Together in Care, as compared to the same amount of months during enrollment. * *p* < 0.05.

**Figure 2 ijerph-19-00458-f002:**
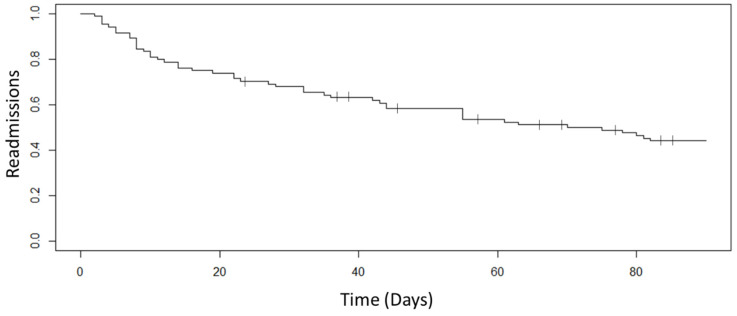
Kaplan–Meier curve for 3-month readmissions of the 84 participants, with censoring provided for the 10 patients who passed away.

**Table 1 ijerph-19-00458-t001:** Characteristics of the participants.

	Patient Population
	*N* = 84
Age (years)	74.9 ± 10.5
Females (%)	33 (39.3)
Body Mass Index	27.0 ± 6.3
Race	
White (%)	65 (77.4)
African American (%)	19 (22.6)
Hispanic/Latino (%)	1 (1.2)
Lives with Family (%)	62 (73.8)
ADI	78.2 ± 17.2
Residing in ADI ≥85 (%)	62 (73.8%)
Sequential Organ Failure Score (at hospital admission)	3.2 ± 1.1
Charlson Co-morbidity Index	7.6 ± 2.5
Co-Morbidities (%)	
COPD	54 (64.3)
Heart Failure	35 (41.7)
Hypertension	72 (85.7)
Diabetes	68 (81.0)
Number of Medications	15.1 ± 6.6
Length of Hospitalization (days)	6.5 ± 4.6
Intensive Care Unit Admission (%)	41 (48.1)
Designated Home Health Care at time of Hospital Discharge	
Nurse (%)	22 (26.2)
PT/OT (%)	10 (8.4)
Physician (%)	1 (1.2)

COPD = Chronic Obstructive Pulmonary Disease; PT/OT = Physical Therapy and Occupational Therapy; ADI = Area Deprivation Index.

**Table 2 ijerph-19-00458-t002:** Key concerns identified during the transition initiative.

Theme	
Insight into Health Condition	“Client wants to be very independent, wants to eat any food, and wants to walk wherever despite his losing balance and falling frequently.”
Trust of the Medical System	“Client states ‘hates going to doctors’ and often times refuses to go to clinic as the client feels ‘they don’t understand me’.”
Access to Health Care Equipment	“Client states she cannot read or write. Unstable living situation and no access to a walker (recommended).”
Competing Priorities	“Client states she cannot afford meds and groceries. Her husband does not work and has no form of income.”
Caregiver Concerns	“Caregiver/daughter feels unable to leave her mother’s side and cannot take a break or leave the house. Client has dementia and is at risk of falling. Client’s daughter cannot afford respite care.”
Communication Access	“Client shares phone with husband, who is usually out of the house. Communication with the client is difficult.”

## Data Availability

Datasets used and/or analyzed during the current study are available from the corresponding author on reasonable requests.

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
