# Peer review of "Together in Care: An Enhanced Meals on Wheels Intervention Designed to Reduce Rehospitalizations among Older Adults with Cardiopulmonary Disease—Preliminary Findings"

_ijerph, 2022, doi:10.3390/ijerph19010458_

Round 1

Reviewer 1 Report

The article presented to me for review raises very important issues in the area of health care and its costs in connection with patient education and support.
The presented pilot studies will certainly inspire the researchers themselves, and perhaps others, to deepen their research in this area.
Let me point out that, in my opinion, the discussion requires supplementation - it lacks the most important elements, i.e. reports from other researchers. There is also a lack of clearly distinguished conclusions that should be found after the discussion.

Author Response

Thank you for the review of our manuscript. Your point on the Discussion section is well taken and we have modified several paragraphs in the Discussion section to point out the reasoning and impact of two of our interventions. Finally, we re-edited the paragraph on our participants and their residing in socioeconomically disadvantaged neighborhoods. All the edits can be tracked.

Reviewer 2 Report

Current repot suggested a finding based upon a partnership between a hospital and a local Meals on Wheels agency to support patients following an incident hospitalization to assess impact on hospital utilization. Please conduct the following concerns.

  1. The transitional care interventions belong to development stage. A new approach to care it is important and useful. Please introduce the main protocol in brief.
  2. The chronic obstructive pulmonary disease (COPD) as secondary outcome needs a rationale because other chronic disorders in aged outcome were also popular.
  3. Medications were not included in the transition team. Why? Aged outcomes were usually received medicines in general.
  4. Evaluation of the transition teams with different health care organizations is the basis of this project.
  5. Insurance company seems like to promote the transitional care interventions particularly using the current results.
  6. A transition team focused on nutrition, home renovations, and social engagement may result in the reduction of re-hospitalizations that needs a potential reason to explain the mechanism.
  7. In the section of conclusion, it seems complicated. Please revise the perspectives in another section before the real findings.

Author Response

Thank you for the feedback. Here are responses, point by point.

  1. The transitional care interventions belong to development stage. A new approach to care it is important and useful. Please introduce the main protocol in brief.  RESPONSE: In 2.2.2The Interventions, we break-up the paragraphs of the intervention, emphasizing all four interventions and their respective time.
  2. The chronic obstructive pulmonary disease (COPD) as secondary outcome needs a rationale because other chronic disorders in aged outcome were also popular. Response: In section 2.4 Outcomes, we added this sentence: The primary outcome focused on hospital expenditures. A subgroup analysis focused on outcomes in patients with COPD, as COPD-related readmissions was the top cause of readmissions in the preceding year. 
  3. Medications were not included in the transition team. Why? Aged outcomes were usually received medicines in general. Response: In Table 1, we do list the amount of medications of each participant (as a mean and standard deviation), as well as mentioned in line 223-225. Apologies if this was not clear.
  4. Evaluation of the transition teams with different health care organizations is the basis of this project. Response: That is correct.
  5. Insurance company seems like to promote the transitional care interventions particularly using the current results. Response: Correct - that would be an ideal outcome, and one that will likely gain attention after our expanded project is complete using these pilot results.
  6. A transition team focused on nutrition, home renovations, and social engagement may result in the reduction of re-hospitalizations that needs a potential reason to explain the mechanism. Response: We spent time in the Discussion section highlighting how home fixtures to prevent falls and social engagement that picked up on symptoms were likely key drivers; further these engagements mitigated socioeconomic factors of the participants who lived in disadvantaged neighborhoods. Overall, these were emphasized in the revised section of the Discussion.
  7. In the section of conclusion, it seems complicated. Please revise the perspectives in another section before the real findings. Response: We have made changes in the Conclusion paragraph by eliminating a sentence that likely adds to more confusion.

Thank you again for these great points.

Reviewer 3 Report

The study presents important contributions to the field of health management and epidemiology applied to health services.

The clipping is original and must be published.

The findings are presented as important for the adoption of intervention measures in health services.

I present some contributions and questions that must be answered point by point:

Abstract:

You wrote: "Rehospitalizations in the Medicare population can be influenced by many social factors,

such as access to food, social isolation and housing security.” Does the term “diverse factors” simply sum up just access to food, social isolation and housing security? There are powerful others in the literature. How about quoting?

Introduction:

Justify the reasons for choosing the primary and secondary outcomes.

Method:

It is very well written and meets the criteria of scientific writing.

Results, discussion and conclusion:

Despite being a study study, the results and discussion are presented as an unfolding of the method very well free. Limitations were found (line 329-340), however, I ask: given the limitations, what are the potentials of the study? Name them.

References:

The study presents important references, however, very old.

I suggest reviewing references older than 10 years (there are many).

Author Response

Thank you so much for the review of our manuscript. Please see our responses to your valuable points:

Abstract:

You wrote: "Rehospitalizations in the Medicare population can be influenced by many social factors,

such as access to food, social isolation and housing security.” Does the term “diverse factors” simply sum up just access to food, social isolation and housing security? There are powerful others in the literature. How about quoting?

Response: Great point and added "diverse" prior to the social factors, while listing the few examples afterwards in preparation for our own intervention.

Introduction:

Justify the reasons for choosing the primary and secondary outcomes.

Response: The primary outcome was chosen as we believed the sample size needed to reduce rehospitalizations was too large for a pilot trial, where we wanted to assure it was feasible. We opted for expenditures as we believed that would be motivation enough to fund a larger project, while expenditures is a reasonable surrogate to hospital-utilization, which is important factor not only for the hospitals but for the patients as well. And as for the second point, I added the justification based on what our hospital sees specifically (meaning, high rates of COPD-related rehospitalizations).

Method:

It is very well written and meets the criteria of scientific writing.

Response: Thank you.

Results, discussion and conclusion:

Despite being a study study, the results and discussion are presented as an unfolding of the method very well free. Limitations were found (line 329-340), however, I ask: given the limitations, what are the potentials of the study? Name them.

Response: Thank you - we have added such potentials at the end of the limitations paragraph.

References:

The study presents important references, however, very old.

I suggest reviewing references older than 10 years (there are many).

Response: Thank you - we have added a few more up to date references: 20-21 and 25-26, both referencing falls and social isolation, respectively.